# Molecular Typing of Pathogenic *Leptospira* Species Isolated from Wild Mammal Reservoirs in Sardinia

**DOI:** 10.3390/ani11041109

**Published:** 2021-04-13

**Authors:** Ivana Piredda, Maria Nicoletta Ponti, Bruna Palmas, Malgorzata Noworol, Aureliana Pedditzi, Lucio Rebechesu, Valentina Chisu

**Affiliations:** Laboratory of Sieroimmunology, Animal Health Department, Istituto Zooprofilattico Sperimentale della Sardegna, 07100 Sassari, Italy; nicoletta.ponti@izs-sardegna.it (M.N.P.); bruna.palmas@izs-sardegna.it (B.P.); malgorzata.noworol@izs-sardegna.it (M.N.); aureliana.pedditzi@izs-sardegna.it (A.P.); lucio.rebechesu@izs-sardegna.it (L.R.); valentina.chisu@izs-sardegna.it (V.C.)

**Keywords:** *Leptospira*, Leptospirosis, mammals, epidemiology, reservoirs, genotyping, MLST

## Abstract

**Simple Summary:**

Leptospirosis is caused by pathogenic spirochaetes of the genus *Leptospira*. Humans can become infected with these bacteria through direct contact with urine from infected animals or indirectly through interaction with a urine contaminated environment. Among wildlife species, rodents are considered the primary reservoir hosts for leptospirosis in rural and urban environments. Epidemiological data, regarding leptospirosis in various wild species in Europe, suggest that these animals play a different role in leptospiral persistence. Unfortunately, studies on the presence and typing of *Leptospira* species in wild mammals are lacking in Sardinia. The aim of the present study was to investigate the prevalence of *Leptospira* species in wild mammals. Kidneys collected from carcasses were analyzed by culture and molecular testing. Greater positivity was found in hedgehogs, followed by weasels and rodents. The results obtained suggest that Sardinian fauna may play a possible sentinel role in the transmission cycle of leptospirosis to humans. Gathering this information in different wildlife species is crucial for better understanding of the epidemiology of leptospirosis and for the development of appropriate prevention measures.

**Abstract:**

Leptospirosis is a global zoonosis caused by pathogenic species of *Leptospira* that infect a large spectrum of domestic and wild animals. This study is the first molecular identification, characterization, and phylogeny of *Leptospira* strains with veterinary and zoonotic impact in Sardinian wild hosts. All samples collected were cultured and analyzed by multiplex real time polymerase chain reaction (qPCR). Sequencing, phylogenetic analyses (based on *rrs* and *secY* sequences), and Multilocus Sequence Typing (MLST) based on the analysis of seven concatenated loci were also performed. Results revealed the detection of *Leptospira* DNA and cultured isolates in 21% and 4% of the samples examined, respectively. Sequence analysis of *Leptospira* positive samples highlighted the presence of the *interrogans* and *borgpetersenii* genospecies that grouped in strongly supported monophyletic clades. MLST analyses identified six different Sequence Types (ST) that clustered in two monophyletic groups specific for *Leptospira*
*interrogans*, and *L. borgpetersenii*. This study provided about the prevalence of leptospires in wild mammals in Sardinia, and increased our knowledge of this pathogen on the island. Monitoring *Leptospira* strains circulating in Sardinia will help clinicians and veterinarians develop strategic plans for the prevention and control of leptospiral infections.

## 1. Introduction

Leptospirosis, which is caused by pathogenic spirochaetes of the genus *Leptospira*, is a re-emerging zoonotic disease with veterinary and public health importance due to its cross-over between humans, domestic animals, and wildlife. Leptospirosis cases have been reported worldwide, in particular in the tropical regions of South and Southeast Asia [1], Africa [2], Western Pacific [3], and Central and North America [4]. It has been estimated that more than one million cases occur each year, including about 60,000 deaths [5].

The most recent advances in the description of *Leptospira* phylogeny have resulted from the study of Vincent et al. [6] in which new species of *Leptospira* belonging to subclades P1 and P2 have been classified on the basis of their pathogenicity. To date, at least 64 different *Leptospira* species have been validated worldwide, based on the average nucleotide identity (ANI) values of their genomes.

The main reservoirs of pathogenic *Leptospira* species are rodents; however, more recently, an increasing number of vertebrate and invertebrate hosts have also been reported to shed this pathogen in their urine. Among rodents, rats represent the main reservoir of pathogenic *Leptospira*, while wild and domestic mammals [7,8], livestock [9,10], amphibians [11], reptiles [12], ticks [13], and bats [14] also appear to play an important role in the spread of the leptospires. Most human infections occur following exposure to soil or water contaminated with the urine of reservoir animals [15].

In Italy, molecular studies have highlighted the presence of pathogenic *Leptospira* in wild boars [16] and pigs [17], porcupines [18], rodents [19,20,21], horses [22], dogs [23], and humans [24]. In Sardinia, recent isolations of *Leptospira* Bratislava and Pomona from wild boars [25] and marine mammals [26] suggest that these animals could potentially act as reservoirs of these serovars. To date, the current literature lacks information on the presence and spread of *Leptospira* species in Sardinian wildlife or the possible role that wild mammals play as maintenance or accidental hosts for these bacteria. The purpose of the present study was to (i) investigate the potential presence of *Leptospira* genomospecies and their sequence types (ST) in wild vertebrates, (ii) identify and genotype leptospires by 16S rRNA, *secY*, and Multilocus Sequence Typing (MLST), and (iii) reconstruct the phylogeny of the obtained sequences.

## 2. Materials and Methods

### 2.1. Sample Collection

Carcasses of small mammals, whose cause of death was related to traffic accidents or attacks from wildlife or domestic animals, were collected between January 2011 to December 2019 from 6 collection sites of North Sardinia (Sassarese, Angola, Gallura, Mantacuto, Nurra, and Goceano). Sardinia, the second largest island in the Mediterranean Sea with an area of 23,821 km^2^, is a region with a great naturalistic importance. The island is characterized by a wide diversity in geology, vegetation and landscape features, surrounded by mountains, forests, and green valleys covered by the typical Mediterranean maquis with Cistus, lentisk, myrtle, and rosemary shrubs. The landscape is also characterized by cultivated coastal plains, watercourses and rocky sheer coasts. Many areas are dedicated to rearing and grazing of sheep, goats, bovines, swine, and horses. Sardinia is also an extraordinary habitat for wild animals such as mouflons, Sardinian deer, wild pigs and foxes, and many birds. The field researchers were trained to sample the dead animals following predetermined guidelines. The carcasses were trans-ported at 4 °C to the laboratory where the general body condition of each carcass was evaluated, and only those animals that did not show obvious signs of deterioration were analyzed. Additionally, nutritional conditions, size, age, and sex of each animal was evaluated before the necropsy. Decomposed carcasses were not included in this study. A total of 387 carcasses were identified phenotypically by an expert veterinarian and included in the study [27]. Dead animals were necropsied and the kidneys were collected under sterile conditions from each animal. Specifically, 25 mg of tissue extracted between the cortical and medullary areas was immediately used for culture, the other was frozen at −20 °C for molecular investigation and characterization of leptospires.

### 2.2. Leptospira spp. Isolation

In order to evaluate the presence or the potential growth of leptospires, collected kidneys were homogenized by using a Stomacher bag (bag filter Avantor^®^, Lutterworth, UK) and inoculated into a sterile tube (Corning—Falcon^®^, Corning, NY, USA) (by using a sterile pipette) containing the commercial semi-solid Ellinghausen-McCullough-Johnson-Harris (EMJH) medium with EMJH enrichment (DifcoTM, BD, Franklin Lakes, NJ, USA), supplemented with 5-Fluorouracil (5-FU; 2 g/L). The media was incubated aerobically at 28 °C and the growth of leptospires was examined under a dark-field microscope weekly, over a period of 3 months for the presence or potential growth of leptospires. Samples that failed to show any evidence of growth after 3 months were considered negative and were discarded [28].

Positive cultures were subjected to purification. Briefly, exceeding nucleotides and primers were inactivated by using the Applied Biosystems™ CleanSweep™ PCR Purification Reagent (Life Technologies Europe BV, Monza MB, Italy), according to vendor’s recommendations. Pure isolates, free of contaminants, were used for serological and molecular identification.

### 2.3. Genomic DNA Extraction

DNA from the kidney was extracted using the DNeasy Blood and Tissue Kit (Qiagen, Hilden, Germany) according to the manufacturer’s instructions. Due to validate the extraction processes and all downstream steps, nuclease-free water and 10 fg of DNA extracted from *Leptospira interrogans* serovar Copenhageni (Fiocruz L1-130) were used as positive and negative controls, respectively. DNA extracted from each sample was stored at −20 °C until use.

### 2.4. Molecular Detection of Leptospira spp. by Multiplex qPCR, and Amplification of rrs and secY Genes

To discriminate between both pathogenic and non-pathogenic leptospires, all DNA samples were tested by multiplex qPCR using lipL32 and 16S rRNA partial target genes. More specifically, primers LipL32-45F (5′-AAGCATTACCGCTTGTGGTG-3′), LipL32-286R (5′-GAA CTCCCA TTT CAG CGA TT-3′), and the probeLipL32-189P (FAM-5′-AA AGC CAG GAC AAG CGCCG-3′-BHQ1) [29], were combined with primers 16S-P1 forward (5′-TAGTGAACGGGATTAGATAC-3′), and 16S-P2 reverse (5′-GGTCTACTTAATCCGTTAGG-3′) and probe 16S-Prob (Cy5-5′-AATCCACGCCCTAAACGTTGTCTAC-3′-BHQ2) that amplify 242 and 104 bp of the *lipL32* and *16S rRNA* genes, respectively. An internal control consisting of exogenous DNA added to the sample before the extraction phase. The qPCR was performed in a total volume of 20 μL containing 50 ng of *Leptospira* spp. genomic DNA, 250 nM of each of the forward and reverse primers, and 10 μL of 5× Master Mix QuantiFast Pathogen PCR + IC Kit (Qiagen, Milan, Italy). All the reactions were performed in duplicates on a 7500 Fast Real-Time PCR System (Applied Biosystems) under the following conditions: 95 °C for 5 min, followed by 45 cycles of denaturation at 95 °C for 15 s, and annealing and elongation for 30 s at 60 °C. A negative control (DNA extracted from water) and a positive control (DNA extracted from the reference strain of *L. interrogans* ATCC^®^ BAA1198D5TM) were included in each PCR test.

Among all positive samples obtained by qPCR, only samples with a threshold cycle (Ct) values lower than or equal to 32 were tested for further analyses. Specifically, 6 kidney samples and 17 *Leptospira* isolates (Table 1) were analyzed with a set of primers that amplified a fragment of 541 bp of the 16S rRNA gene, and of 549 bp of the *secY* partial gene [30]. Negative and positive controls were included in each test, with a negative and a positive for every 20 samples tested. The PCR reactions were performed by using a T100 Thermal Cycler (Bio-Rad apparatus). PCR products were visualized by electrophoresis in 1.5% agarose gel stained with SYBR-Safe DNA Gel Stain (Invitrogen, Carlsbad, CA, USA), and examined under UV transillumination.

### 2.5. Sequencing and Phylogenetic Analyses

All *rrs* (16S rRNA gene) and *secY* positive amplicons were purified and directly sequenced by using an BigDye terminator cycle sequencing ready reaction kit (Life Technologies, UK). Sequences were edited with Chromas 2.2 (Technelysium, Helensvale, Australia), then aligned with Clustal X [31] in order to assign them to unique sequence types, and checked against the GenBank database with nucleotide blast (BLASTn) [32]. Multiple sequence alignments and sequence similarities were calculated using the Clustal W [33] and the identity matrix options of BioEdit [34], respectively. For phylogenetic analyses, the sequence types obtained in this study were aligned with a set of 22 sequences representing *rrs* and *secY* variability of the different species belonging to the genus *Leptospira*.

### 2.6. MLST Analysis of Leptospira Isolated Strains

In order to reveal Sequence Types (ST) of *Leptospira* isolates, MLST was performed using 7 housekeeping genes: *pntA*, *sucA*, *tpiA*, *pfkB*, *mreA*, *glmU,* and *caiB* [35]. Each allele and the allelic profiles (glmU-pntA-sucA-tpiA-pfkB-mreA-caiB) were submitted to the *Leptospira* database (http://pubmlst.org/leptospira, accessed on January 2021) to define the STs. Phylogenetic analysis was performed using MEGA6 software [36].

## 3. Results

### 3.1. Detection of Leptospira Exposure and Infection in Wild Mammals

A total of 387 carcasses of 15 different animal species belonging to Rodentia (n = 177 46%; 95% CI: 41–51%), Erinaceomorpha (n = 37 10%; 95% CI: 7–13%), Carnivora (n = 162 42%; 95% CI: 37–47%), and Lagomorpha (n = 11 3%; 95% CI: 1–5%) orders, were collected in this study (Table 2). The majority of the samples analyzed were adults (n = 340 88%; 95% CI: 85–91%). All samples did not show any macroscopic lesions compatible with *Leptospira* infection after pathologic examination post mortem. The 387 kidneys tested, 80 (21%; 95% CI 17–25%) samples belonging to 7 animal species, were positive for pathogenic *Leptospira* species upon amplification by using multiplex qPCR. Bacterial cultures revealed that 4% (n = 17/387; 95% CI: 2–6%) of the kidney samples were positive for *Leptospira* approximately after 60 days of incubation. All kidney cultures isolates exhibited *Leptospira* positivity after qPCR analyses. The results of qPCR and cultures are presented in Table 2.

### 3.2. Characterization of Leptospira Isolates

Sequencing results performed on the *rrs* and *secY* amplicons obtained from the 17 kidney cultures and the 6 samples (23 positive samples in total) from qPCR analyses produced clear sequencing signals, with an identity superior to 99% (Table 3). Among the *rrs* positive samples, the resulting BLASTn analysis revealed that 9 sequences were members of *L. borgpetersenii* group, 13 belonged to the *L. interrogans* group, and 1 sequence exhibited the highest homology with the intermediate *L. johnsonii* (100% identity). Samples positive for the *Leptospira* species by using 16S rRNA target gene were also positive when tested with the set of primers targeting the protein translocase *secY* subunit present in *Leptospira* species. Sequencing of the 23 *secY* amplicons revealed that 14 (61%; 95% CI: 41–81%) and 9 (39%; 95% CI: 19–59%) sequences were 99–100% similar to *L. interrogans* and *L. borgpetersenii* strains, respectively. The *rrs* and *secY* sequence types, the host origin of all sequences and BLASTn identity are shown in Table 3.

The MLST analysis of the 23 *Leptospira* strains, allowed to obtain 6 different sequence types (ST), belonging to ST149 (derived from 3 *Apodemus sylvaticus*, 1 *Rattus rattus*, 4 *Erinaceus europaeus*, and 1 *Martes martes*), ST198 (found in 1 *Apodemus sylvaticus*, 4 *Erinaceus europaeus*, 1 *Martes martes,* and 1 *Vulpes vulpes*). ST36 (from 1 *Rattus rattus* and 1 *Rattus norvegicus*), ST24 (from 1 *Mustela nivalis*), ST17 (from 1 *Rattus rattus*), and ST140 (from 1 *Vulpes vulpes*). MLST based on 7-loci scheme results obtained from the 23 *Leptospira* isolates are shown in Table 4.

### 3.3. Phylogenetic Analysis

Phylogenetic analysis based on the alignment of the 14 rrs sequence types obtained in this study with the 22 *Leptospira* reference sequences (Figure 1), identified 3 main groups representative of the pathogenic, intermediate, and saprophytic *Leptospira* species.

More specifically, sequence types named Seqrrs2, Seqrrs3, Seqrrs4, Seqrrs5, Seqrrs6, Seqrrs7, Seqrrs8, Seqrrs9 Seqrrs10, Seqrrs11, Seqrrs12, and Seqrrs13 grouped in a strongly supported clade with pathogenic *Leptospira* strains while Seqrrs1 was included in a separate clade including the intermediate *Leptospira* species. The phylogenetic trees obtained by aligning the six *secY* sequence types and the 6 STs resulted from MLST analyses with the *Leptospira* reference strains, indicated that all isolates from this study grouped with reference strains representative of *L. interrogans* and *L. borgpetersenii*, respectively. It indicated that all sequences here detected can be classified within the pathogenic *Leptospira* group. The sequence clusters obtained were statistically supported by bootstrap analyses (Figure 2 and Figure 3).

## 4. Discussion

The importance of rodents as reservoirs for a variety of *Leptospira* serovars has been widely studied in the world [37]. There is an increasing interest in monitoring of *Leptospira* spp. hosts, and studies on the prevalence of this pathogen in wild mammals are increasing over Europe. In Germany, 6% of animals tested positive for *L. kirschneri* and *L. interrogans* [38]. In France, studies conducted on reported serological positivity for *L. interrogans*, *L. kirschneri*, and *L. borgpetersenii* in 24 different mammalian species [39]. In Sardinia, studies on presence and typing of *Leptospira* species in wild mammals are still lacking.

This report examined circulating *Leptospira* strains in 15 different wild species, including rodents, using culture and DNA characterization tools. Our results indicate that all wild species examined are carriers of pathogenic *Leptospira* species in Sardinia. Pathogenic *Leptospira* were found with a frequency of 54% (95% CI: 38–70%) in hedgehogs, followed by mustelids with 40% (95% CI: 0–83%), end wild rodents with 21% (95% CI: 10–33%). The detection of *L. interrogans* serovar Australis and *L. borgpetersenii* serovar Ballum in hedgehogs was in agreement with other studies conducted in several European countries, including France [40], Italy [41], the Netherlands [42], and Scotland [43], which showed the presence of these *Leptospira* spp. from Erinaceomorphs. Moreover, pathogenic *Leptospira* were isolated from hedgehogs in France, as well as in China [44] recently. These findings show that hedgehogs could act as important source of pathogenic *Leptospira* spp. serogroup Australis and outline the importance of leptospirosis surveillance in this species. In this study, the presence of *L. interrogans* in the carnivore group was demonstrated for the first time on the island. Among the species analyzed was *Vulpes vulpes ichnusae*, a Sardinian species endemic to urban and peri-urban areas, including human environments. The hunting of this species is allowed on the island, and is regulated by the Regional Law number 23 of 1998. Foxes are known to be an important source of *Leptospira* in Europe, where the Grippotyphosa serogroup is known as the most frequently reported in Germany [45], and the Poi and Saxkoebing serogroups, and Sejroe are the most common in Poland [46]. The presence of pathogenic *Leptospira* in Sardinian wild foxes needs further investigation to clarify the role of wild carnivores as a reservoir of pathogenic *Leptospira* serovars on the island, as well as their epidemiological role in the zoonotic cycle. Still within the carnivore group, we also report the first molecular detection of *L. borgpetersenii* and *L. interrogans* from two species of Sardinian Mustela and one Martens. Together with the hedgehog and the fox, it is among the mammals most commonly hit by cars. Additionally, for these species the results obtained are in agreement with the studies conducted in France which show that the mustelid species have the highest risk of being infected by *L. interrogans*, *L. borgpetersenii,* and *L. kirschneri* [40]. This study also reveals the presence of *L. interrogans* in Sardinian rodents (*Rattus norvegicus*, *Rattus rattus,* and *Apodemus sylvaticus*). It has been known that wild rats (*Rattus* spp.) Are the most important sources of *Leptospira* infection, as they are abundant in urban and peri-domestic environments [39]. The brown rat is reported to be the primary host of *L. interrogans* related to the serogroup Icterohaemorrhagiae, which is responsible for the most severe forms of the disease in humans [47,48]. With the identification of 14 different *rrs* sequences, 6 for the *secY* gene, and 6 MLST profiles, our study reflects the wide diversity of *Leptospira* genotypes circulating in wildlife. Only 6 samples from biological matrix were added to the molecular analysis, the remaining 74 positive sample for *Leptospira* could not be amplified, most likely due to low DNA concentrations.

In the present study, the use of the *rrs* and *secY* genes represented a useful tool for detecting the *Leptospira* genomospecies in wild mammals and allows differentiating members of the pathogenic, intermediate and saprophytic *Leptospira* group. The 3 types obtained showed an obvious phylogenetic divergence from all recognized species belonging to the 3 clades of *Leptospira* and it was in accordance with previous studies [6,49]. Results obtained by using *rrs* and *secY* genes are the same as those obtained by MLST analyses except for those obtained from 1 *Mustela nivalis*. The partial ribosomal 16S gene sequence detected in this mammal hosts, was identical to that of a species already isolated from soil in Japan namely *L. johnsonii* and belonging to the intermediate clade [50]. However, the presence of this P2 intermediate species were not confirmed with *secY* and MLST analyses, indicating that the molecular identification of *Leptospira* strains needs the use of further genetic markers in order to confirm these results.

## 5. Conclusions

Our results show that several pathogenic strains of *Leptospira* are circulating in Sardinian fauna. Information on the possible role as sentinel or reservoirs of wild mammals is critical to understand the possible zoonotic potential of *Leptospira*. Therefore, the characterization of the genetic diversity of *Leptospira* strains is fundamental to designing epidemiological studies and control strategies for leptospirosis in the same area. Further studies are needed to better characterize isolates by analyzing more discriminative genes, and to identify the main reservoirs of *Leptospira* strains in Sardinia island.

## Figures and Tables

**Figure 1 animals-11-01109-f001:**
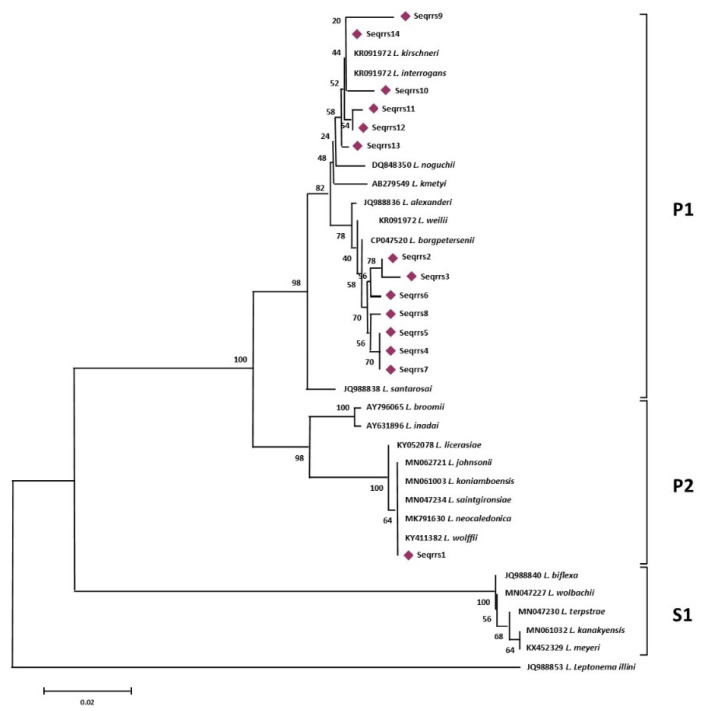
*rrs*-based phylogenetic analyses of the sequence types generated in this study and of 22 sequences representative of the different species of the genus *Leptospira*. Evolutionary history was inferred using the Neighbor-Joining method. The optimal tree with the sum of branch length = 0.41105704 is shown. The percentage of replicate trees in which the associated taxa clustered together in the bootstrap test (1000 replicates) are shown next to the branches.

**Figure 2 animals-11-01109-f002:**
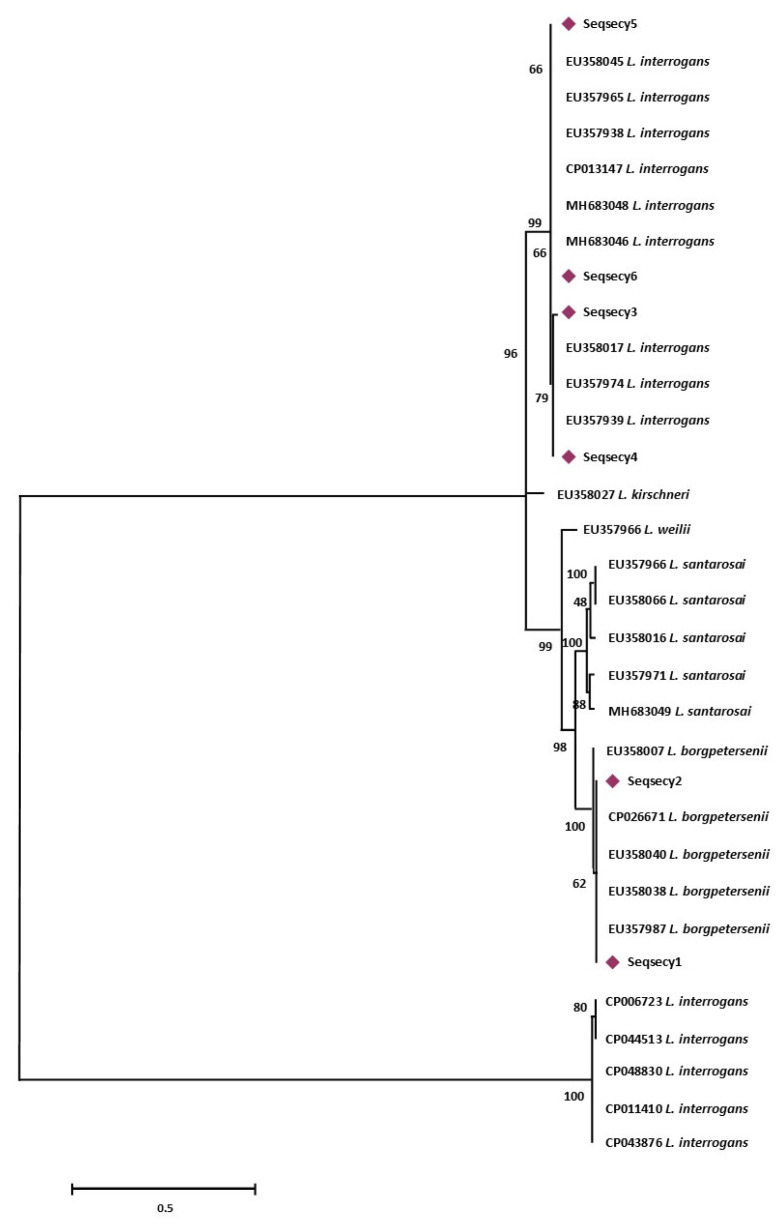
Maximum Likelihood method based on the Tamura 3-parameter model. The analysis involved 32 nucleotide sequences. All positions containing gaps and missing data were eliminated. There were a total of 486 positions in the final dataset. Statistical support for internal branches of tree was evaluated by bootstrapping with 1000 reiterations.

**Figure 3 animals-11-01109-f003:**
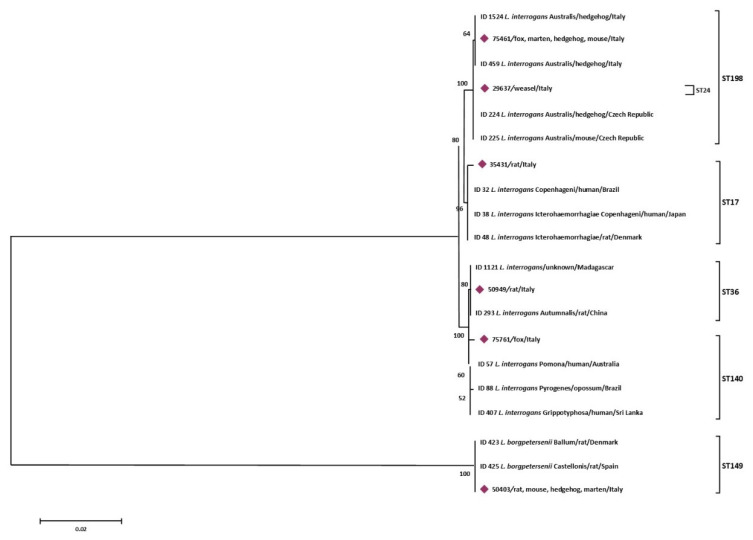
Phylogenetic tree based on concatenated sequences of 7 housekeeping loci of the 6 STs obtained in this study was constructed by using the Maximum Likelihood method based on the Tamura 3-parameter model. The tree with the highest log likelihood (−6579,6990) is shown. The percentage of trees in which the associated taxa clustered together is shown next to the branches. The analysis involved 20 nucleotide sequences. All positions containing gaps and missing data were eliminated. There were a total of 3108 positions in the final dataset.

**Table 1 animals-11-01109-t001:** Origin of samples investigated in this study.

Host Species (*Scientific Name*)	Source	Isolate ID
Wild mouse *(Apodemus sylvaticus)*	culture	105080
	culture	109472
	culture	112931
	culture	89848
Brown rat *(Rattus rattus)*	culture	50403
	culture	50949
	kidneys	35431
Norway rat *(Rattus norvegicus)*	culture	58418
	culture	59153
European hedgehog *(Erinaceus europaeus)*	culture	42608
	culture	78309
	culture	23218
	culture	79986
	culture	79987
	kidneys	73267
	kidneys	85389
	kidneys	17536
	kidneys	45527
Weasel *(Mustela nivalis boccamela)*	culture	29637
Marten *(Martes martes latinorum)*	culture	3517
	kidneys	41982
Red Fox *(Vulpes vulpes ichnusae)*	culture	64874
	culture	75461
	culture	75761

**Table 2 animals-11-01109-t002:** Percentages of *Leptospira* spp. isolated from kidney samples from multiplex qPCR and culture from different host species in Sardinia.

Groups	Species	No. of Carcasses (♂;♀)	qPCR (%; 95%CI))	POS Culture (%; 95%CI)
Rodents	*Apodemus sylvaticus*	60 (41;19)	19/60 (32; 20–44)	4/60 (6.7; 1–13)
	*Mus musculus*	1 (1;0)	0/1	0/1
	*Crocidura russula ichnusae*	4 (4;0)	0/4	0/4
	*Suncus etruscus pachyurus*	3 (2;1)	0/3	0/3
	*Eliomys quercinus sardus*	1 (0;1)	0/1	0/1
	*Myocastor coypus*	11 (5;6)	0/11	0/11
	*Rattus rattus*	53 (24;29)	11/53 (21; 10–33)	2/53 (4; 0–9)
	*Rattus norvegicus*	44 (24;20)	7/44 (16; 4–28)	1/44 (2; 0–6)
Erinaceomorphs	*Erinaceus europaeus*	37 (15;22)	20/37 (54; 38–70)	5/37 (13; 3–25)
Large Carnivores	*Mustela nivalis boccamela*	5 (5;0)	2/5 (40; 0–83)	1/5 (20; 15–55)
	*Martes martes latinorum*	32 (17;15)	6/32 (19; 5–33)	1/32 (3; 0–9)
	*Mustela vison domestica*	1 (0;1)	0/1	0/1
	*Vulpes vulpes ichnusae*	124 (90;34)	15/124 (12; 6–18)	3/24 (2; 0–19)
Lagomorphs	*Oryctolagus cuniculus*	3 (2;1)	0/3	0/3
	*Lepus capensis mediterraneus*	8 (2;6)	0/8	0/8
Total		387 (232;155)	80/387 (21; 17–25)	17/387 (4; 2–6)

**Table 3 animals-11-01109-t003:** Designation, sequence type, and maximum identities of the *rrs* and *secY* gene sequence types identified in this study.

rrs Sequence Types	ID	Origin (n.)	BLASTn Analyses (Identity)
Seqrrs1	29637	*Mustela nivalis boccamela* (1)	*L. johnsonii* (100%)
Seqrrs2	109472	*Apodemus sylvaticus* (1)	*L. borgpetersenii* (99–100%)
Seqrrs3	112931	*Apodemus sylvaticus* (1)	
Seqrrs4	79987; 85389	*Erinaceus europaeus* (2)	
Seqrrs5	3517	*Martes martes latinorum* (1)	
Seqrrs6	105080	*Apodemus sylvaticus* (1)	
Seqrrs7	23218	*Erinaceus europaeus* (1)	
Seqrrs8	78309; 50403	*Erinaceus europaeus* (1) *Rattus rattus* (1)	
Seqrrs9	42608;17536; 45527;73267; 41982	*Erinaceus europaeus* (4) *Martes martes latinorum* (1)	*L. interrogans* (99–100%)
Seqrrs10	64874	*Vulpes vulpes ichnusae* (1)	
Seqrrs11	50949	*Rattus rattus* (1)	
Seqrrs12	75761; 59153; 35431	*Vulpes vulpes ichnusae* (1) *Rattus norvegicus* (1) *Rattus rattus* (1)	
Seqrrs13	89848	*Apodemus sylvaticus* (1)	
Seqrrs14	79986; 75461	*Erinaceus europaeus* (1) *Vulpes vulpes ichnusae* (1)	
**secY Sequence Types**	**ID**	**Origin (n.)**	**BLASTn Analyses (Identity)**
SeqsecY1	112931; 109472; 50403; 85389; 79987; 3517	*Apodemus sylvaticus* (2) *Rattus rattus* (1) *Erinaceus europaeus* (2) *Martes martes latinorum* (1)	*L. borgpetersenii* (99–100%)
SeqsecY2	23218; 78309; 105080	*Erinaceus europaeus* (2) *Apodemus sylvaticus* (1)	
SeqsecY3	50949; 59153	*Rattus rattus* (1) *Rattus norvegicus* (1)	*L. interrogans* (99–100%)
SeqsecY4	75761; 64874; 73267; 79986; 42608	*Vulpes vulpes ichnusae* (2) *Erinaceus europaeus* (3)	
SeqsecY5	29637; 41982; 75461; 17536; 45527; 89848	*Mustela nivalis boccamela* (1) *Martes martes latinorum* (1) *Vulpes vulpes ichnusae* (1) *Erinaceus europaeus* (2) *Apodemus sylvaticus* (1)	
SeqsecY6	35431	*Rattus rattus* (1)	

**Table 4 animals-11-01109-t004:** Numbers of alleles and sequence types (ST) of 23 pathogenic *Leptospira* strains.

Animal Species	ID	Allelic Profile MLST							STs (*Leptospira* species)
		*glmU*	*pntA*	*sucA*	*tpiA*	*pfkB*	*mreA*	*caiB*	
*Apodemus* *sylvaticus*	105080	24	32	30	36	67	26	12	149 (*L. borpetersenii* Ballum)
	109472	24	32	30	36	67	26	12	149 (*L. borpetersenii* Ballum)
	112931	24	32	30	36	67	26	12	149 (*L. borpetersenii* Ballum)
	89848	1	66	2	1	5	3	4	198 (*L. interrogans* Australis)
*Rattus rattus*	50403	24	32	30	36	67	26	12	149 (*L. borpetersenii* Ballum)
	50949	3	2	3	3	4	5	5	36 (*L. interrogans* Autumnalis)
*Rattus norvegicus*	35431	1	1	2	2	10	4	8	17 (*L. interrogans* Icterohaemorrhagiae)
	59153	3	2	3	3	4	5	5	36 (*L. interrogans* Autumnalis)
*Erinaceus* *europaeus*	42608	1	66	2	1	5	3	4	198 (*L. interrogans* Australis)
	78309	24	32	30	36	67	26	12	149 (*L. borpetersenii* Ballum)
	23218	24	32	30	36	67	26	12	149 (*L. borpetersenii* Ballum)
	79986	1	66	2	1	5	3	4	198 (*L. interrogans* Australis)
	79987	24	32	30	36	67	26	12	149 (*L. borpetersenii* Ballum)
	73267	1	66	2	1	5	3	4	198 (*L. interrogans* Australis)
	85389	24	32	30	36	67	26	12	149 (*L. borpetersenii* Ballum)
	17536	1	66	2	1	5	3	4	198 (*L. interrogans* Australis)
	45527	1	66	2	1	5	3	4	198 (*L. interrogans* Australis)
*Mustela nivalis boccamela*	29637	1	4	2	1	5	3	4	24 (*L. interrogans* Australis)
*Martes martes* *latinorum*	3517	24	32	30	36	67	26	12	149 (*L. borpetersenii* Ballum)
	41982	1	66	2	1	5	3	4	198 (*L. interrogans* Australis)
*Vulpes vulpes* *ichnusae*	64874	1	66	2	1	5	3	4	198 (*L. interrogans* Australis)
	75461	1	66	2	1	5	3	4	198 (*L. interrogans* Australis)
	75761	3	3	3	3	4	5	16	140 (*L. interrogans* Pomona)

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
