# Peer review of "Molecular Typing of Pathogenic Leptospira Species Isolated from Wild Mammal Reservoirs in Sardinia"

_animals, 2021, doi:10.3390/ani11041109_

Round 1
Reviewer 1 Report
Piredda and colleages performed a molecular investigation of pathognenic leptospira carried by Sardinian wildlife.
While the study carried out here is not very innovative and is similar to what has already been done in some other countries, their work is very complete from the point of view of the temporal sampling (over 8 years) and the number of analyzed individuals (387)
Moreover, the authors have used adequate methodologies to carry out their study. The material and methods section is correctly described with the adequate level of information to understand the experiments performed.
Regarding the results, the reviewer finds them very well presented and correctly supporting the conclusions made at the end of the study.
The reviewer regrets, however, that the detection of L. Johnsonii (a species of the P2 clade formerly known as intermediates) in one animal is not discussed further. L. Johnsonii was isolated in Japan from a soil sample and the fact that this P2 species was found in a mammal deserves to be discussed further.
Furthermore, have the authors ever performed a molecular investigation of leptospires species present in the environment in Sardinia? The presence of L. Johnsonii (and other pathogenic leptospira) in the soils of some areas of Sardinia could help to better understand the epidemiology of Leptospirosis in Sardinia.
Concerning the nomenclature of Leptospira species, I recommend to the authors to adopt the new nomenclature recently published (Vincent et al 2019) and to correct the legend of figure 1 by replacing pathogenic by S1 , intermediate by S2 and Saprophytes by S1.
Finally the reviewer noted some typo errors:
Material and method section, 2.2 subsection, line 2: ul should be µl to be consistent throughout the manuscript.
Result section, subsection 3.2, line 2 : explane should be explained
Author Response
Point 1: The reviewer regrets, however, that the detection of L. Johnsonii (a species of the P2 clade formerly known as intermediates) in one animal is not discussed further. L. Johnsonii was isolated in Japan from a soil sample and the fact that this P2 species was found in a mammal deserves to be discussed further.
Response 1: The detection of L. johnsonii has been discussed in L 311-316 that now reads:
"The partial ribosomal 16S gene sequence detected in this mammal host, was identical to that of a species already isolated from soil in Japan namely L. johnsonii and belonging to the intermediate clade (Masuzawa et al., 2019). However, the presence of this P2 intermediate species were not confirmed with secY and MLST analyses indicating that the molecular identification of Leptospira strains needs the use of further genetic markers in order to confirm these results”.
Point 2: Furthermore, have the authors ever performed a molecular investigation of leptospires species present in the environment in Sardinia? The presence of L. Johnsonii (and other pathogenic leptospira) in the soils of some areas of Sardinia could help to better understand the epidemiology of Leptospirosis in Sardinia.
Response 2: These data will be part of a distinct paper, which will include a greater number of samples coming from different geographical regions.
Point 3: Concerning the nomenclature of Leptospira species, I recommend to the authors to adopt the new nomenclature recently published (Vincent et al 2019) and to correct the legend of figure 1 by replacing pathogenic by S1 , intermediate by S2 and Saprophytes by S1.
Response 3: The nomenclature has been corrected according to Vincent et al.
Point 4: Material and method section, 2.2 subsection, line 2: ul should be µl to be consistent throughout the manuscript.
Response 4: According to your suggestion, ul was replaced with µl
Point 5: Result section, subsection 3.2, line 2 : explane should be explained
Response 5: The right verbal form has been corrected
Reviewer 2 Report
A nicely written cross-sectional study to characterize the prevalenc of Leptospira in Sardinia.
Most of my comments are tracked in the attached pdf
Additional references, as there are many, in lines 53-55 to further support widespread nature of leptospirosis would be helpful.
Please inlcude your specific hypotheses. This serves to guide the reader as to what you plan set out to answer. This also would help you identify appropriate statistcial tests to answer necessary questions (see below).
Methods:
Please include more information regarding sample area- size, population density
Statistcial tests should be used to answer some questions that your important work revelated. I added 95% CI to some sections, but they should be included throughtout (including Tables). Univariate statistics to compare prevalence between species is important to characterize highest prevalence and species most likely ot be reservoirs. Also, univariate statistics could be performed on findigs recording during exam as they relate to leptospira status: sex, nutritional condition, size, and age. Are there differences in Lepto status by sex, age, location found, and nutritional status (eg., if no difference in body condition than may presume the anmals are not infected but reservoirs). Follow-up logistics regression could be done if univariate statistcis are significant. Statistcis could likewise be performed on molecular results.
Performing statistics wil expand your discussion and allow more direct comparison to other populations regarding epidemiology of Lepto on this island.

Author Response
Point 1: Additional references, as there are many, in lines 53-55 to further support widespread nature of leptospirosis would be helpful.
Response 1: Additional references has been added in L 54-56
Point 2: Please inlcude your specific hypotheses. This serves to guide the reader as to what you plan set out to answer. This also would help you identify appropriate statistcial tests to answer necessary questions (see below).
Point 3: Statistcial tests should be used to answer some questions that your important work revelated. I added 95% CI to some sections, but they should be included throughtout (including Tables). Univariate statistics to compare prevalence between species is important to characterize highest prevalence and species most likely ot be reservoirs. Also, univariate statistics could be performed on findigs recording during exam as they relate to leptospira status: sex, nutritional condition, size, and age. Are there differences in Lepto status by sex, age, location found, and nutritional status (eg., if no difference in body condition than may presume the anmals are not infected but reservoirs). Follow-up logistics regression could be done if univariate statistcis are significant. Statistcis could likewise be performed on molecular results.
Response 2 and 3: More information about sex, nutritional condition, and age of the mammals here collected are now present in the text. However, we believe this is a preliminary report in which we described for the first time the presence of Leptospira species in wild mammals from North-Sardinia. Making general assumptions with this small numbers collected from a limitated area of the island could lead to conclusions not statistically supported by data. Although, this is an interesting point which we will discuss in a next article including a larger number of samples collected from the whole island. However, 95% CI has been added throughout the paper. In addition, the study area was added in M&M section.
Point 4: Please include more information regarding sample area- size, population density
Response 4: Additional infomations has been added in L 75-87
Reviewer 3 Report
In the manuscript entitled “Molecular typing of pathogenic Leptospira species isolated from wild mammal reservoirs in Sardinia”, the authors aim to investigate, by molecular and microbiological techniques, the presence of Leptospira species in the Sardinian wild fauna. The manuscript can be of interest to the field, but some issues should be first addressed.
--------
M&M
- L94: Did the authors include rabbit serum, in-house prepared supplements, or commercial Leptospira enrichment to the EMJH medium??
-L98: The sentence is redundant: ‘weekly’ and ‘every seven days’.
-L100: Please describe the ‘purification’ procedure.
-Genomic DNA extraction and PCR: Did the authors include positive controls, for example, tissue from experimentally infected animals?
-L115-117: ligA/B and lipL32 are different Leptospira genes. It is very strange that the oligonucleotide ligA/B identifies lipL32 gene. Please indicate whether the target was ligA, ligB and/or lipL32 genes.
-Please provide the sequence of the oligonucleotides used to amplify the target genes.
-Table 1: The authors could include the common name of the animals to facilitate the unspecialized audience’s understanding.
-L144: Indicate the meaning of ‘rrs’.
--------
Results
-Is the positivity for Leptospira presence consistent between culture and PCR analysis? i.e. the same samples positive in culture also positive in PCR.
-Do the authors have information on which genes were amplified in each sample? Were the multiplex PCR probes labeled with different dyes?
- Are the animals positive for Leptospira considered chronic asymptomatic carriers or are they susceptible to leptospirosis (become ill)?
- The authors should mention if any signs or macroscopic lesions compatible with leptospirosis were found in the post-mortem analysis. Were there lesions compatible with other known and identifiable diseases? Were the animals considered healthy?
- It would be very interesting to see histologic images of the organs in which Leptospira DNA was detected. Both to analyze possible microscopic lesions and visualize Leptospira or leptospiral antigens (immunohistochemistry, for example).
--------
Discussion
-L251: substantial leptospires
-L258: The authors should provide the full list of the 23 animal species included in the study. Tables 1 and 3 mention 7 species, and table 2 mentions 15 species.
-5-Fluorouracil is usual in the isolation medium for Leptospira. However, it is genotoxic and the isolates could show modifications. Do the authors think that this could have exerted a significant impact on the results? Please discuss.
-------
- The manuscript needs an English revision to correct grammar and semantic issues.
Author Response
M&M
Point 1: - L94: Did the authors include rabbit serum, in-house prepared supplements, or commercial Leptospira enrichment to the EMJH medium??
Response 1: The information about the DIFCO Leptospira medium with EMJH enrichment has been added in M&M L102-105
Point 2: -L98: The sentence is redundant: ‘weekly’ and ‘every seven days’.
Response 2: The sentence now reads: “The media was incubated aerobically at 28°C and the growth of leptospires was examined weekly under a dark-field microscope”. L105-107
Point 3: -L100: Please describe the ‘purification’ procedure.
Response 3: The purification procedure used has been added in L109-112
Point 4: -Genomic DNA extraction and PCR: Did the authors include positive controls, for example, tissue from experimentally infected animals?
Response 4: Positive and negative extraction controls have been used in our study and are now inserted in L118-120
Point 5: -L115-117: ligA/B and lipL32 are different Leptospira genes. It is very strange that the oligonucleotide ligA/B identifies lipL32 gene. Please indicate whether the target was ligA, ligB and/or lipL32 genes.
Response 5: All information requested were added in L128-135
Point 6: -Please provide the sequence of the oligonucleotides used to amplify the target genes
Response 6: The sequence of the oligonucleotides used to amplify the target genes were added in L129-134
Point 7: -Table 1: The authors could include the common name of the animals to facilitate the unspecialized audience’s understanding.
Response 7: All information requested were added in Table 1
Point 8: -L144: Indicate the meaning of ‘rrs’.
Response 8: It has been added
Results
Point 1: -Is the positivity for Leptospira presence consistent between culture and PCR analysis? i.e. the same samples positive in culture also positive in PCR.
Response 1: this information has been added as follows: “All kidney cultures isolates exhibited Leptospira positivity after qPCR analyses.” L190-191
Point 2: -Do the authors have information on which genes were amplified in each sample? Were the multiplex PCR probes labeled with different dyes?
Response 2: The set of primers of multiplex qPCR were labeled with different dyes and they have been used for a screening of Leptospira species in collected samples. However, a standard PCR by using species-specific primers (secy and rrs) were then used due to identify all Leptospira genospecies. This information is reported in L129-134 and L145-148
Point 3: - Are the animals positive for Leptospira considered chronic asymptomatic carriers or are they susceptible to leptospirosis (become ill)?
Response 3: Since macroscopic examination of kidneys did not allow to identify any lesion compatible with Leptospira infection, we could postulate that animals examined can be considered as asymptomatic carriers. Other types of studies will highlight if wild mammals are reservoir or susceptible of Leptospira infection in the island.
Point 4: - The authors should mention if any signs or macroscopic lesions compatible with leptospirosis were found in the post-mortem analysis. Were there lesions compatible with other known and identifiable diseases? Were the animals considered healthy?
Response 4: See above
Point 5: - It would be very interesting to see histologic images of the organs in which Leptospira DNA was detected. Both to analyze possible microscopic lesions and visualize Leptospira or leptospiral antigens (immunohistochemistry, for example).
Response 5: Focus of this paper was the molecular investigation of Leptospira pathogens in wild mammals, therefore we did not analyse possible microscopic lesions and visualized leptospiral antigens that could represent and interesting point for future studies
Discussion
Point 1: -L251: substantial leptospires
Response 1: Indeed, the sentence lacks clarity and now reads: “Our results indicate that all species examined could be carriers of pathogenic Leptospira species in Sardinia." L269-270
Point 2: -L258: The authors should provide the full list of the 23 animal species included in the study. Tables 1 and 3 mention 7 species, and table 2 mentions 15 species.
Response 2: The sentence lacked clarity, and it has been reformulated in L268-269: “This study examined circulating Leptospira strains in 15 different wild species including rodents, using DNA characterization tools.”
Point 3: -5-Fluorouracil is usual in the isolation medium for Leptospira. However, it is genotoxic and the isolates could show modifications. Do the authors think that this could have exerted a significant impact on the results? Please discuss.
Response 3: The aim of this study was to identify the presence of Leptospira spp. in wild mammals for the first time in Sardinia. The EMJH medium supplemented with 0.1% of 5-Fluorouracil is usually used for Leptospira isolation and allows to decrease the vitality of the other microorganisms that could inhibit the growth of Leptospira. Other studies on genotoxic effects of the 5-Fluorouracil in Leptospira isolates will be performed in the future.
Point 4: - The manuscript needs an English revision to correct grammar and semantic issues.
Response 4: The manuscript has been edited and subjected to careful revision
